# Synthetic rEg.P29 Peptides Induce Protective Immune Responses Against *Echinococcus granulosus* in Mice

**DOI:** 10.3390/vaccines13030266

**Published:** 2025-03-03

**Authors:** Yongxue Lv, Jing Tang, Tao Li, Yinqi Zhao, Changyou Wu, Wei Zhao

**Affiliations:** 1School of Basic Medicine, Ningxia Medical University, Yinchuan 750004, China; yongxuelv2021@163.com (Y.L.); tangjing0514@yeah.net (J.T.); 2Ningxia Key Laboratory of Prevention and Control of Common Infectious Diseases, Ningxia Hui Autonomous Region, Yinchuan 750004, China; zhaoyinqi2024@163.com; 3Department of Hepatobiliary Surgery, General Hospital of Ningxia Medical University, Yinchuan 750004, China; longtaodaxia123@126.com; 4Institute of Immunology, Zhongshan School of Medicine, Sun Yat-sen University, Guangzhou 5102275, China; changyou.wu@outlook.com

**Keywords:** *Echinococcus granulosus*, rEg.P29, synthetic peptides vaccine, protective effect

## Abstract

**Background:** *Echinococcus granulosus* represents a significant threat to animal husbandry and human health, but its consequences are often underestimated. Vaccination can prevent *E. granulosus* infection. We investigated the immune protective effect induced by the recombinant protein P29 of *E. granulosus* (rEg.P29) peptide vaccine. **Methods**: The CD4^+^ T-, CD8^+^ T-, Treg-, and CD8^+^CD107a^+^ T-cell proportions in the spleen and peripheral blood of infected mice were analyzed using flow cytometry. Additionally, we measured the proportions of IFN-γ and IL-2 secreted by memory T cells, CD19^+^CD138^−^B cells, CD19^+^CD138^+^ plasmablasts, CD19^−^CD138^+^ plasma cells, and CD19^+^IgD^−^IgG^+^ and CD19^+^IgD^−^IgA^+^ memory B cells. **Results**: No significant differences were noted in CD4^+^ T-, CD8^+^ T-, and CD8^+^CD107a^+^ Treg-cell percentages among the experimental groups. However, IFN-γ, IL-2, and TNF-α levels and vaccine-specific antibody concentrations in the plasma were significantly elevated in the rEg.P29_T+B_ + CpG + infection and rEg.P29 + CpG + infection groups compared to those in the PBS + infection and CpG + infection groups. Similarly, CD19^−^CD138^+^ plasma cell and CD19^+^IgD^−^IgG^+^ and CD19^+^IgD^−^IgA^+^ memory B-cell populations, along with specific antibodies, were significantly higher in these groups. Especially, the average cyst burden in the rEg.P29_T+B_ + CpG + infection and rEg.P29 + CpG + infection groups was significantly reduced compared to that in the PBS + infection and CpG + infection groups. **Conclusions**: Synthetic peptide vaccines targeting rEg.P29 can effectively inhibit cysts, offering a novel strategy for the development of vaccines against *E. granulosus*. These findings provide a foundation for further research on the immunogenicity and protective efficacy of rEg.P29-based vaccines.

## 1. Introduction

Echinococcosis is a zoonotic parasitic disease caused by the larval stage of *Echinococcus* tapeworms in humans and animals with a global distribution [1,2,3]. Eggs exposed to the external environment exhibited strong tolerance and viability. Adult parasites reside in the small intestines of carnivorous animals such as dogs and wolves, while intermediate hosts include even-toed hoofed animals such as sheep and primates (including humans) [4,5]. Statistically, approximately 4 million people are infected with hepatic hydatid disease globally, and an additional 60 million are at risk of infection. This situation seriously jeopardizes global public health security and economic development [6]. Echinococcosis causes economic losses to herders as livestock die and infected organs are discarded. It also disables infected patients, drives many into or back into poverty, and challenges local public health. Despite strengthened prevention and control efforts everywhere, the situation remains severe owing to complex transmission routes and a wide range of susceptible people.

Contemporary therapeutic strategies rely predominantly on albendazole, praziquantel, and invasive surgical interventions. However, these approaches face substantial limitations, such as suboptimal efficacy, risks of drug resistance, and environmental contamination from persistent anthelmintic residues [7,8,9]. Surgical management, although necessary for advanced cases, carries a significant financial burden and is associated with postoperative complications. These challenges underscore the critical need for prophylactic strategies aligned with the WHO’s “prevention-first” paradigm for parasitic disease control [10,11].

However, the development of anthelmintic vaccines poses a constant and formidable challenge owing to host-induced evasion mechanisms triggered by the parasitic process of the worm. These evasion mechanisms significantly complicate vaccine development [12,13,14]. Recent advances in vaccine research have been achieved through technological developments. Synthetic peptide vaccines, such as those synthesized artificially without nucleic acid components, offer potential solutions for the limitations associated with conventional vaccines. Consequently, they are considered more efficient, stable, and cost-effective alternatives [15,16,17,18]. rEg.P29 has been previously isolated and demonstrated robust immune protection in sheep [19,20]. To enhance the safety and efficacy of vaccine development, we screened rEg.P29 peptides for T- and B-cell epitopes and successfully synthesized a peptide vaccine for rEg.P29 [21,22]. Furthermore, this peptide vaccine primarily induces T helper type 1 (Th1)-mediated cellular immune responses and IgG-based antibody responses [23], in contrast to previous attempts using Eg95-derived peptides, which failed to induce protective immunity despite antibody generation [24,25].

This discrepancy highlights the importance of epitope selection strategies and adjuvant compatibility in subunit vaccine design. To address the translational gap between immunogenicity and functional protection, we conducted a longitudinal challenge study evaluating the efficacy of the synthetic rEg.P29 peptide against *E*. *granulosus* infection in C57BL/6 mice. Through comprehensive analysis of cyst burden quantification, cytokine profiling (Th1/Th2), and antibody isotyping (IgG1/IgG2a) at 5 months post-infection, this investigation aims to: (1) validate the vaccine’s prophylactic potential and (2) establish a framework for optimizing peptide-based antiparasitic vaccines.

## 2. Materials and Methods

### 2.1. Antigen and Vaccines

Synthetic peptides based on the peptide analysis of rEg.P29 were synthesized by Sangon Biotech (Shanghai, China). To guarantee specificity, peptides used for enzyme-linked immunosorbent assay (ELISA) and in vitro stimulation did not contain Keyhole Limpet Hemocyanin (KLH). KLH is often used as a carrier protein that binds to small-molecule antigens to enhance their immunogenicity of the latter. Peptides coupled with KLH were employed to immunize mice (T-cell and B-cell peptide-coupled KLH carrier [rEg.P29_T+B_]), as previously reported [23]. Expression and purification of rEg.P29 were performed as previously described. Briefly, expression was induced for 12 h in the presence of 50 μg/mL isopropyl-β-D-thiogalactoside (IPTG; Invitrogen, Carlsbad, CA, USA). The rEg.P29 protein was purified using the His Purification Kit (Merck, Kenilworth, NJ, USA) [19].

### 2.2. Experimental Protocol

C57BL/6 mice (aged 6–8 weeks) were obtained from the animal center of Ningxia Medical University. After one week of acclimatization, the animals were randomly divided into four groups of five individuals each. All experiments were approved by the Ethics Committee of Ningxia Medical University. CpG ODN1826 was used as an adjuvant and synthesized by Sangon Biotech (Shanghai, China). The experimental groups were as follows: PBS + infection group (injected with PBS only), CpG group + infection (injected with 40 μg CpG only), rEg.P29_T+B_ + CpG + infection group (injected with 20 μg rEg.P29_T+B_ and 40 μg CpG), and rEg.P29 + CpG + infection group (injected with 20 μg rEg.P29 and 40 μg CpG). Immunization was administered subcutaneously to the abdominal region. Using a prime-boost strategy, immunization was carried out thrice successively, as illustrated in Figure 1A.

Protoscoleces were isolated from the cysts of the patients, and their morphology and vitality were observed under a microscope (hydatid cysts were provided by the General Hospital of Ningxia Medical University). To guarantee the consistency and precision of the experiment, the protoscoleces employed for infection were sourced from the same patient. Another part was taken and stained with eosin to determine its vitality, as depicted in Figure 1B. After 4 weeks post-immunization, the protoscoleces were resuspended in PBS, and 1000 protoscoleces were injected intraperitoneally into the mice in all four groups.

As shown in Figure 1D, all groups of mice survived 5 months after infection. Anesthetized mice were subjected to ocular extraction to obtain peripheral blood, which was collected in heparinized anticoagulant tubes. The anticoagulated tube was then centrifuged at 450× *g* for 5 min to isolate plasma. Cysts were separated from the abdominal cavity of the mice and placed in pre-weighed 1.5 mL tubes for reweighing. The spleens were separated and transferred into designated tubes. Mice from all four groups were dissected, and the weight of the hydatid cysts was measured following 5 months of protoscoleces infection.

### 2.3. Lymphocyte Preparation and Culture

Five months post-infection, the mice were anesthetized. Subsequently, splenic and peripheral blood lymphocytes were isolated to assess the effects of T cells. Peripheral blood from the mice was suspended in Hank’s balanced salt solution and isolated using Ficoll–Hypaque density gradient centrifugation (Tianjin HaoYang Biological Manufacture, Tianjin, China) at 350× *g* for 20 min. Spleen tissues were mechanically disrupted and filtered through a 70 μm cell strainer. Subsequently, splenic lymphocytes were isolated using Ficoll–Hypaque. The lymphocytes were collected and washed twice with Hank’s balanced salt solution before being resuspended at a final concentration of 1 × 10^6^ cells/mL in complete RPMI 1640 medium (HyClone, Logan, UT, USA) supplemented with 10% heat-inactivated fetal calf serum (FCS; geminiBio, West Sacramento, CA, USA), streptomycin at a concentration of 100 μg/mL, penicillin at a concentration of 100 U/mL, L-glutamine at a concentration of 2 mM, and 2-mercaptoethanol at a concentration of 50 μM (Gibco, Grand Island, NE, USA).

### 2.4. ELISA and ELISPOT Assay

ELISA was conducted to determine the cytokine level. The levels of cytokines in ex vivo-isolated single cells from the spleens of infected mice which were quantified using commercial ELISA kits, following the manufacturer’s instructions. The cells were stimulated with 15 μg/mL of rEg.P29_T+B_ or rEg.P29 for 72 h at 37 °C. The culture supernatant was then analyzed using ELISA to detect IFN-γ, IL-2, TNF-α, IL-4, and IL-17A. All ELISA kits were purchased from BD Biosciences, except for IL-17A (Mabtech, Nacka, Sweden).

For the enzyme-linked immune spot assay (ELISPOT) of cytokine-producing cells, the splenocytes were stimulated with 15 μg/mL of rEg.P29_T+B_ or rEg.P29 for 24 h at 37 °C and 5% CO_2_. The frequencies of splenocytes producing IFN-γ or IL-4 were measured using mouse IFN-γ/IL-4 ELISPOT kits (BD Biosciences, San Jose, CA, USA) according to the manufacturer’s instructions. Spot-forming cells (SFCs) were enumerated using an automated ELISPOT plate reader (Aid Autoimmun Diagnostika Gmbh, Straßberg, Germany).

### 2.5. Flow Cytometry and Intracellular Cytokine Staining

Flow cytometry (FCM) was conducted to quantify immunocytes in the peripheral blood and spleen lymphocytes. The cells were washed twice with PBS containing 0.1% bovine serum albumin and 0.05% sodium (Buffer 1). The cells were stained with the LIVE/DEADTM Fixable Viability Dye eFluorTM 510 (TermoFisher, Waltham, MA, USA). In simple terms, for surface FACS staining, cells were washed twice with Buffer 1, followed by staining with fluorochrome-conjugated monoclonal antibodies for phenotyping for 30 min at 4 °C in the dark. Then, the cells were washed with Buffer 1 and filtered for detection. For intracellular cytokine staining, single-cell suspensions (1 × 10^6^ cells) from control or infected mice were stimulated with rEg.P29_T+B_ or rEg.P29 (15 μg/mL) at 37 °C for 20 h. Brefeldin A (Sigma-Aldrich, St Louis, MO, USA) was added to the culture during the final 6 h at a concentration of 10 µg/mL. After surface staining, the cells were washed with Buffer 1, fixed with 4% paraformaldehyde, and permeabilized with Buffer 2 (Buffer 1 with 0.1% saponin) overnight at 4 °C. Intracellularly stained with anti-IFN-γ and anti-IL-2 at 4 °C for 30 min, cells were washed with Buffer 2 and evaluated using the FACSCelesta (BD Biosciences, San Jose, CA, USA) for data collection. Data were analyzed using FlowJo 10 software (TreeStar, San Carlos, CA, USA). The fluorochrome-conjugated monoclonal antibodies are listed in Table 1.

### 2.6. Detection of Specific Antibody Response with ELISA

First, the ELISA plates were coated with rEg.P29_T+B_ or rEg.P29 (10 μg/mL) in coating buffer overnight at 4 °C. The plate was washed with PBS containing 0.05% Tween 20 (PBST) and blocked with PBST containing 5% skim milk at 37 °C for 1 h and washed by PBST. Then, after adding the plasma of mice from each group, the ELISA plate was incubated at 37 °C for 2 h and was washed by PBST. The Goat anti-Mouse IgM, IgG, IgA, and IgE (Abcam, Cambridge, UK) secondary antibody that was labeled with horse radish peroxidase was incubated at 37 °C for 1 h. After washing the ELISA plate again, 3,3′,5,5′-Tetramethylbenzidine (TMB) was added, and the plate was incubated for color development. The OD450nm values were measured using an ELISA plate reader.

Ninety-six well plates were coated with 10 μg/mL rEg.P29_T+B_ or rEg.P29, and antigen-specific IgG1 and IgG2a (Abcam, Cambridge, UK) responses in the serum of mice were measured as previously described (the mice blood serum was diluted in decimal dilution method).

### 2.7. Statistical Analysis

All statistical analyses were performed using the GraphPad Prism 10.0 (GraphPad Software, Inc., La Jolla, CA, USA). For comparisons between the two groups, an unpaired Student’s *t*-test was used. When comparing more than two groups, one-way or two-way analysis of variance was used. The data are presented as the mean ± standard deviation, with a sample size of n = 5. A *p*-value > 0.05 was considered not significant.

## 3. Results

### 3.1. Evaluation of Potential Protective Effects of Designed Vaccines

To verify the potential protective effects of the vaccine designed in this study, we established an animal model of infection using mice challenged with protoscoleces. The infected mice were monitored for 5 months for body weight changes and death. Additionally, as shown in Figure 1C, all mice gained body weight steadily, and there was no significant difference in body weight among the four groups. During the observation period, no mice died in any group (Figure 1D). Furthermore, cyst weight was also measured as an intuitive indicator for evaluating the resistance of *E. granulosus*. The results showed that there was no statistical difference in cyst weight between the immunized and non-immunized groups, as depicted in Figure 1E. However, it is noteworthy that the vaccine-immunized group exhibited an inhibitory effect on the cysts. The average cyst weight of immunized mice was less than that of non-immunized mice, which indicated that the vaccine had a good protective effect and could prevent the infection of *E. granulosus* protoscoleces properly. It was hypothesized that the CpG + infection group could potentially induce non-specific immune responses. Therefore, the CpG + infection group was designated as the control group. Subsequently, the cyst weight less rates of the rEg.P29_T+B_ + CpG + infection group and rEg.P29 + CpG + infection group were calculated. These rates were 82% and 80%, respectively (Figure 1F).

### 3.2. Evaluation of Vaccine-Induced T-Cell Immune Response

FCM was employed to assess the proportions of CD4^+^ and CD8^+^ T cells, revealing no significant differences in these cell populations between the immunized and non-immunized groups (Figure 2A–E). Furthermore, there were no differences in regulatory T cells (Tregs) in the spleen and peripheral blood samples from mice between the two groups (Figure 2F–H).

Subsequently, we conducted the ability of the vaccine to elicit T-cell mediated immune response following exposure to protoscoleces and quantified the release of IFN-γ, IL-2, TNF-α, IL-4, and IL-17A using ELISA. Compared with the PBS + infection and CpG + infection groups, both the rEg.P29_T+B_ + CpG + infection and rEg.P29 + CpG + infection groups exhibited significantly higher levels of IFN-γ, IL-2 and TNF-α. However, there were no significant differences in the levels of IL-4 and IL-17A between the groups (Figure 3A–E). Simultaneously, lymphocytes were collected for IFN-γ/IL-4 ELISPOT assay using rEg.P29_T+B_ or rEg.P29 as stimuli. In comparison to the PBS + infected group, immunization with rEg.P29_T+B_ or rEg.P29 resulted in a higher number of specific IFN-γ-expressing cells. Conversely, all groups showed similar numbers of IL-4-expressing cells (Figure 3F,G).

A previous study indicated long-lasting protection of immunization-induced anti-*E. granulosus* immunity is associated with a sustained Th1 immune response following the protoscoleces challenge. We investigated the production of cytokines by memory T cells 5 months after infection. Here, we used the production of IL-2 and IFN-γ as indicators of cellular immune response, as shown in Figure 4A–F. The proportion of IL-2 and IFN-γ secreted by CD4^+^CD44^+^ and CD8^+^CD44^+^ T cells in the vaccine immunization group exhibited a significant increase compared to the PBS + infection and CpG + infection group.

The surface expression of CD107a is a reliable measure of cytolytic capacity. CD107a is a vesicle membrane protein of cytolytic granules. It expresses on the surface of effector cytotoxic T lymphocytes during degranulation. As shown in Figure 4G–I, CD107a expression in the spleen of mice was only higher in the rEg.P29 + CpG + infection group than in the CpG + infection group, and there was no difference in CD107a expression in the peripheral blood between the groups, indicating that the cytotoxicity of T cells in the rEg.P29 + CpG + infection group was significantly enhanced 5 months after infection.

### 3.3. Evaluation of Vaccine-Induced Humoral Immunoreaction

In addition to detecting memory T cells, we also detected plasma cells, plasmablasts, and B cells (Figure 5A). After comparing different types of B cells, we found that the proportions of CD19^+^CD138^−^B cells and CD19^+^CD138^+^ plasmablasts were higher in the PBS + infection group than in the rEg.P29_T+B_ + CpG + infection and rEg.P29 + CpG + infection group (Figure 5B,C). Furthermore, the proportion of CD19^−^CD138^+^ plasma cells in the PBS + infection group was lower than those in the rEg.P29_T+B_ + CpG + infection and rEg.P29 + CpG + infection group (Figure 5D). Additionally, the proportions of CD19^+^IgD^−^IgA^+^ and CD19^+^IgD^−^IgG^+^ memory B cells in the rEg.P29 + CpG group were higher than those in the PBS + infection group; however, there were no differences among the remaining groups (Figure 5E–G). The antibody response was detected 5 months after infection. ELISA was used to detect anti-rEg.P29_T+B_/rEg.P29 specific IgM, IgG, IgA, and IgE antibodies. The results demonstrated that both the rEg.P29_T+B_ + CpG + infection group and the rEg.P29 + CpG + infection group were capable of predominantly eliciting IgM and IgG antibody responses specific to rEg.P29 (Figure 6A,B). Finally, specific IgG1 and IgG2a antibody titers were detected using the terminal titration method. The results showed that the IgG1 titers of the rEg.P29_T+B_ + CpG + infection group and rEg.P29 + CpG + infection group were slightly lower than that of the IgG2a antibody (Figure 6C,D).

## 4. Discussion

Cestode infections caused by *E. granulosus* constitute a formidable challenge to global livestock production systems while simultaneously posing substantial zoonotic threats to public health. Despite its substantial socioeconomic impact, the disease burden associated with helminthiasis remains underappreciated in many endemic regions [26]. Recent epidemiological data revealed an increasing rise in *E. granulosus* prevalence, which correlates with exacerbated economic losses in pastoral communities and increased human morbidity [27,28]. Among the emerging immunoprophylactic candidates, rEg.P29 has demonstrated promising immunogenic potential through host-protective response induction. However, considering the ongoing genetic optimization requirements for rEg.P29-based vaccine formulations, there is a critical need to develop novel, cost-effective vaccination strategies that balance immunological efficacy with practical implementation feasibility.

Previous investigations have demonstrated that the rEg.P29_T+B_ peptide vaccine induces a Th1-based cellular immune response and an IgG-based antibody response [23], yet its definitive protective efficacy remained uncharacterized. Our experimental design employed synthetic peptide vaccination in murine models followed by controlled protoscoleces challenge, revealing significant clinical protection evidenced by stable somatic growth trajectories, absence of pathognomonic symptoms, and complete survival maintenance. In this study, we employed a synthetic peptide vaccine to immunize mice followed by infection with protoscoleces, resulting in stable weight gain without clinical symptoms or mortality. Although no statistically significant difference in cyst weight existed among the groups, an inhibitory effect was still evident in terms of the cyst reduction rate. The cyst weight inhibition rate of the rEg.P29_T+B_ + CpG + infection group was nearly identical to that of the rEg.P29 + CpG + infection group, which is of significant importance. We believe that the host’s immune response is complex. The inhibitory effect of the vaccinated group on protoscoleces may have manifested mainly through immune-related mechanisms. For example, specific immune cells are activated to attack the protoscoleces and impair their function. However, this does not necessarily lead to an immediate change in cyst weight. The host immune system may establish a local equilibrium state, which not only restricts the excessive growth of protoscoleces but also prevents a rapid decline in cyst weight.

The immunoevasive strategies of *E. granulosus* principally involve metacestode-mediated host immunomodulation, necessitating vaccine-induced rebalancing of the Th1/Th2 axis for effective parasitocidal activity [29,30]. Successfully treated patients with *E. granulosus* exhibit low levels of IL-10, absence of IL-4, and high levels of IFN-γ, while elevated levels of IL-10, IL-4, and reduced levels of IFN-γ are observed [31]. Our immunological profiling through quantitative ELISA and ELISPOT assays confirmed vaccine-mediated modulation of splenic lymphocyte cytokine secretion patterns, particularly enhancing Th1-associated interferon-gamma production while attenuating Th2-related interleukin expression. Furthermore, Tregs play a crucial role in regulating host resistance to parasite infection through complex cellular and molecular mechanisms [14,32]. However, there was no significant difference in the number of Tregs between the groups in this study.

Immune-inducing memory cells located at the site of pathogen invasion must elicit prompt and effective responses. Vaccine-induced peripheral circulating antibodies and memory cells can effectively prevent the occurrence and progression of diseases [33]. Therefore, establishing long-term resident CD4^+^ and CD8^+^ memory T cells, as well as memory B cells in barrier tissues through local immunization to enhance regional immunity, is pivotal for vaccine design. CD44^+^CD4^+^ and CD44^+^CD8^+^ T cells can secrete IL-2 and IFN-γ, with CD8^+^T cells potentially promoting activation of CD4^+^T cells through cytokine secretion, thereby exerting anti-infective effects against viral infections, intracellular bacteria, parasites, etc. [34,35]. However, the role of CD8^+^ T cells in antiparasitic infection studies has often been overlooked. In addition to cytokine secretion, CD8^+^T cells eliminate target cells by releasing granzyme and perforin. The expression level of the surface marker CD107a in activated cytotoxic lymphocytes reflects the extent of degranulation in the CD8^+^T cell population [36,37,38]. The present study aimed to assess the proportion of CD8^+^CD107a^+^ T cells derived from the mouse spleen and peripheral blood. Our findings revealed that the rEg.P29 + CpG + infection group had a significantly higher proportion of cells than the CpG + infection group. Additionally, the rEg.P29_T+B_ + CpG + infection group was slightly higher than the CpG + infection group, but the difference was not statistically significant.

Humoral immunoreactivity is important for antiworm immunity. Our results showed that the proportion of CD19^−^CD138^+^ plasma cells and CD19^+^IgD^−^IgA^+^ and CD19^+^IgD^−^IgG^+^ memory B cells in the rEg.P29 + CpG + infection group was higher than that in the PBS + infection group. Additionally, our ELISA results indicated that the rEg.P29_T+B_ peptide vaccine induced high levels of anti-rEg. P29 and anti-rEg.P29_T+B_ cell-specific antibodies. Previous studies have shown that the antibody- and complement-mediated clearance of invasive protocols is the dominant host protection mechanism induced by anti-*E. granulosus* vaccines [39,40]. A previous study showed that higher OD values at 450 nm were associated with higher levels of protection [41]. In this study, rEg.P29 or rEg.P29_T+B_ peptide vaccines were infected 5 months after immunization, and the levels of anti-rEg. P29 or anti-rEg.P29_T+B_ specific antibodies induced by the rEg.P29_T+B_ peptide vaccine in mice were higher than those in the PBS + infection or CpG + infection groups. Our results showed that rEg.P29_T+B_ specific antibodies also have anti-infective effects.

Studies have shown that the IgG2a/IgG1 ratio is associated with protective immune response [42,43]. Our results revealed that anti-rEg. P29 and rEg.P29_T+B_ specific IgG2a exhibited slightly higher titers compared to IgG1. The finding suggested that murine IgG2a may have protective properties, which is also supported by our data. The functional role of IgA in *E. granulosus* infection remains unclear.

## 5. Conclusions

We have demonstrated that the rEgP29_T+B_ peptide vaccine can inhibit the growth of cysts. This study has laid a solid foundation for further development of a multi-peptide vaccine and acceleration in the progress towards an anti-*E. granulosus* vaccine. However, this study still has some limitations, we have not verified whether the coupling ratio of T-cell and B-cell epitopes affects the inhibition rate of the vaccine against cysts. This is what needs to be verified in our subsequent research.

## Figures and Tables

**Figure 1 vaccines-13-00266-f001:**
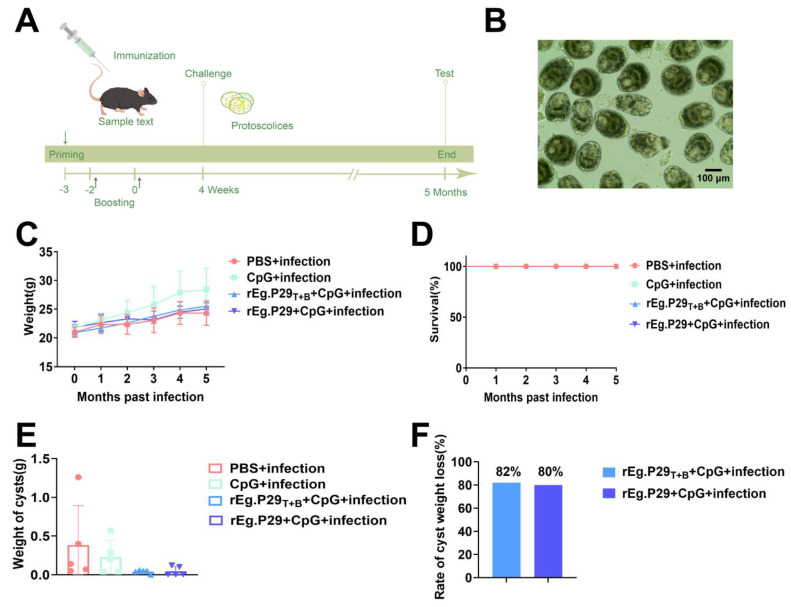
Immune protection of rEg.P29 peptide vaccine in mice. (**A**) The procedure of immunization, protoscolices challenge and sample collection in mice. Female C57BL/6 mice (6–8 weeks of age) were inoculated with rEg.P29 peptide vaccine, CpG or mock-injected with PBS, respectively. Samples were collected at 5 months post-immunization. Protoscolices challenge was carried out at 4 weeks post-immunization. (**B**) Microscopic picture of protoscolices. Body weights of immunized mice were monitored monthly for a period of 5 months to evaluate the pathogenicity of rEg.P29 peptide vaccine (**C**) The body weights of the mice were recorded. (**D**) The survival rates were recorded. (**E**) The cyst weight of mice in each group. (**F**) The cyst weight less rates.

**Figure 2 vaccines-13-00266-f002:**
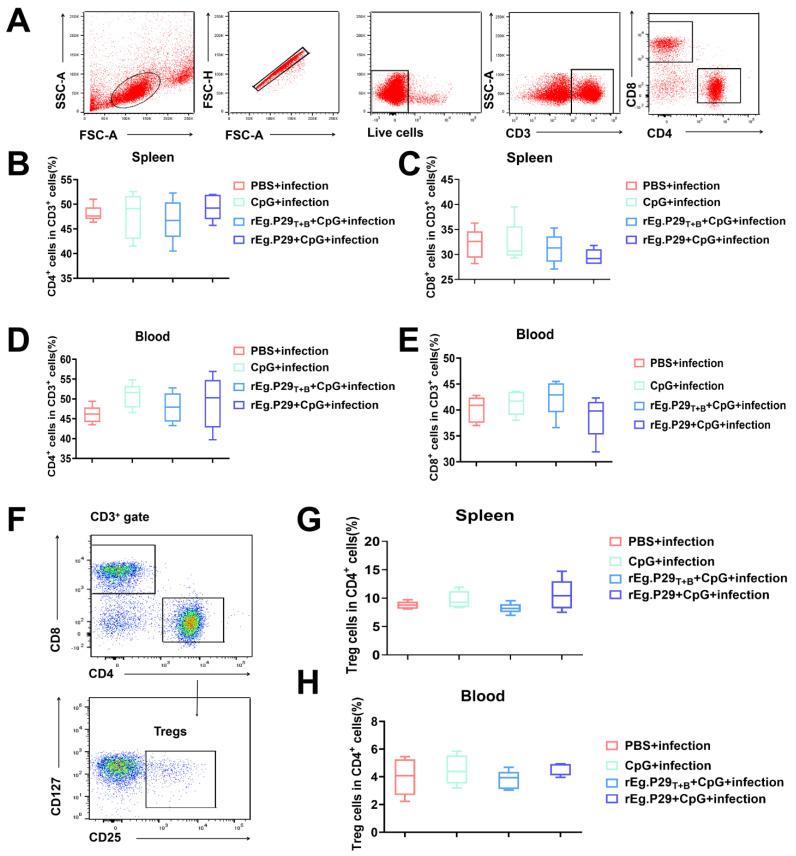
The specific cellular immunity tested by FCM. (**A**) The gating strategies for analyzing the CD4^+^ and CD8^+^T cells. (**B**–**E**) The percentage of CD3^+^CD4^+^and CD3^+^ CD8^+^ T cells of every groups. (**F**) The gating strategies for analyzing the Tregs. (**G**,**H**) The percentage of Tregs in the mice of every groups.

**Figure 3 vaccines-13-00266-f003:**
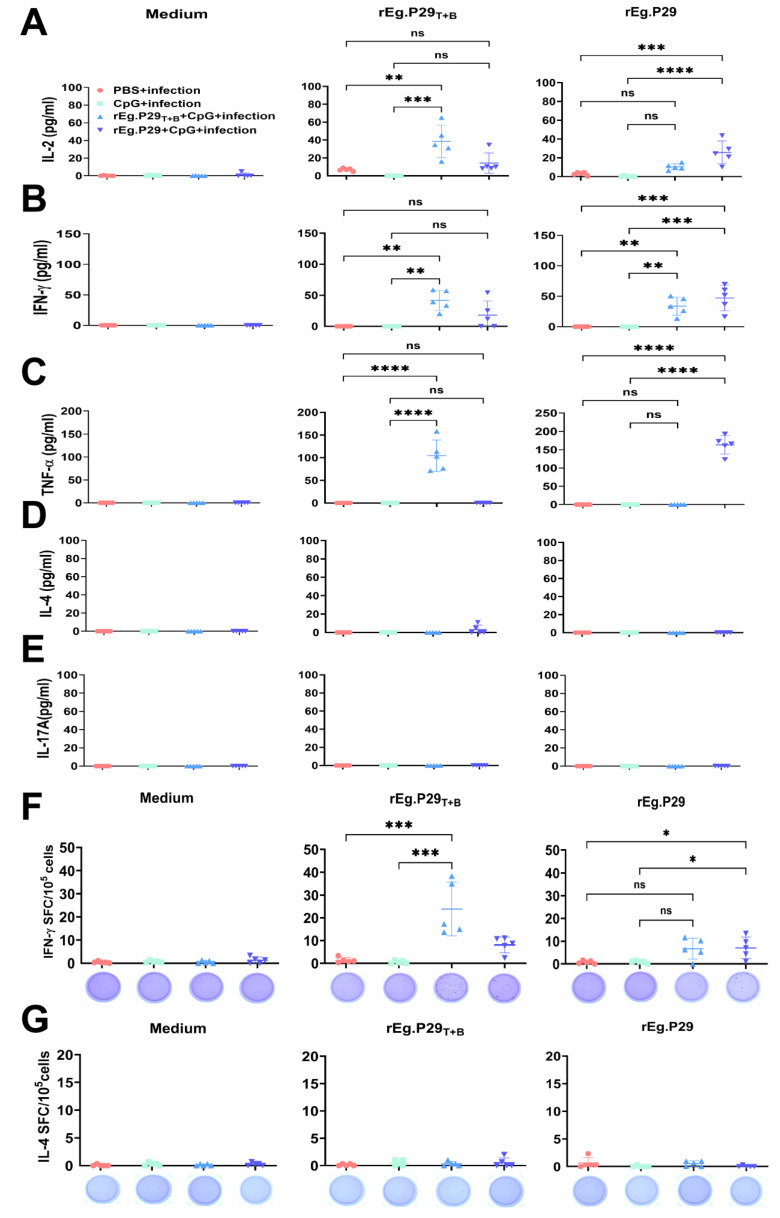
The specific cellular immunity tested by ELISA and ELISPOT. (**A**–**E**) The supernatants were tested for IL-2, IFN-γ, TNF-α, IL-4 and IL-17A productions by ELISA. The secretion of antigen-specific IFN-γ and IL-4 in splenocytes were measured using ELISPOT kits. (**F**) Antigen-specific IFN-γ spot-forming cells (SFCs). (**G**) Antigen-specific IL-4 SFCs. Representative images of IFN-γ or IL-4 SFCs are shown below each graph. * *p* < 0.05, ** *p* < 0.01, *** *p* < 0.001, **** *p* < 0.0001, ns, not significant.

**Figure 4 vaccines-13-00266-f004:**
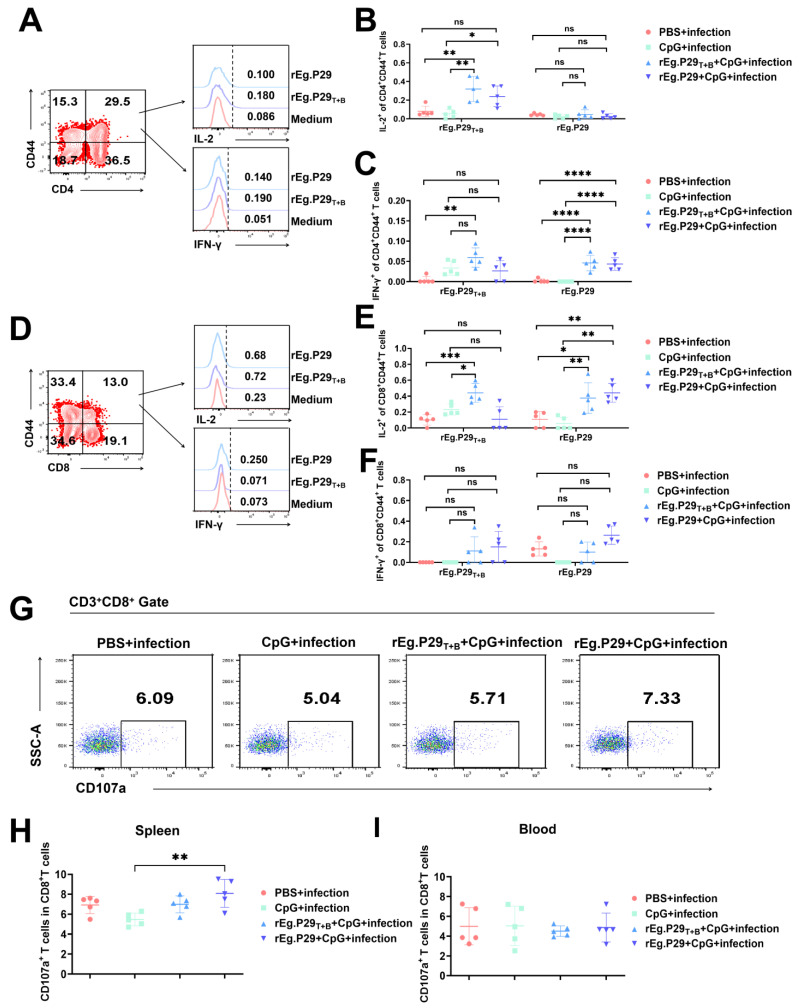
Detection of cytokine secreted by memory T cells and CD8^+^CD107a^+^T cells tested by FCM. (**A**) Representative flow cytometric peak-plots of CD4^+^CD44^+^ T cells were shown. Analyses for (**B**,**C**) IL-2 and IFN-γ from the CD4^+^CD44^+^T cells in spleen. (**D**) Representative flow cytometric peak-plots of CD8^+^CD44^+^ T cells were shown. Analyses for (**E**,**F**) IL-2 and IFN-γ from the CD8^+^CD44^+^T cells in spleen. Single cell suspensions were prepared to detect CD8^+^CD107a^+^ T cells by FCM. (**G**) Representative flow cytometric dots of CD8^+^CD107a^+^ T cells in the mice splenocytes of every groups were shown. (**H**,**I**) The quantified data of CD8^+^CD107a^+^ T cells from mice. * *p* < 0.05, ** *p* < 0.01, *** *p* < 0.001, **** *p* < 0.0001, ns, not significant.

**Figure 5 vaccines-13-00266-f005:**
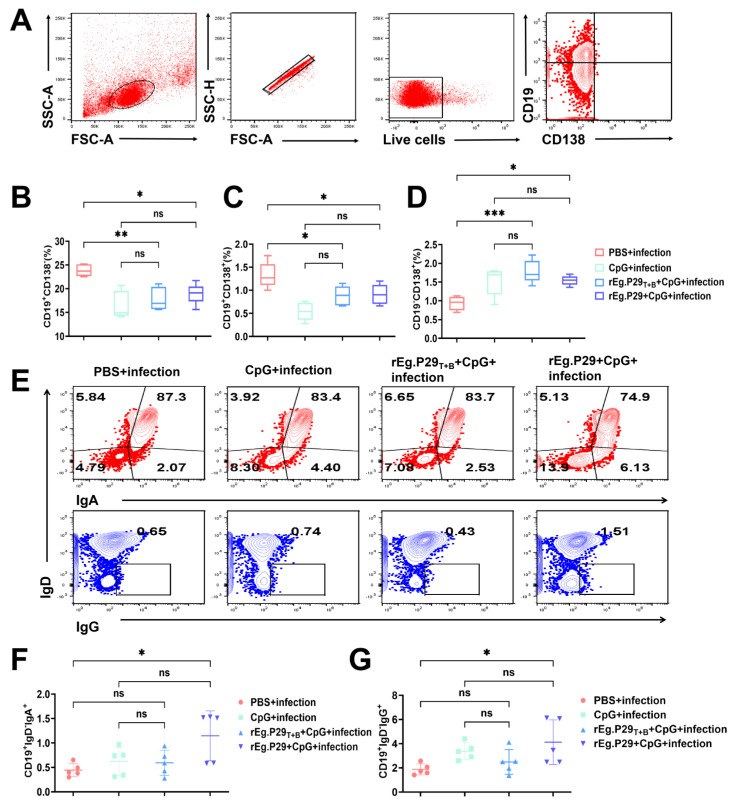
Cytokine secretion profiling in plasma cells, plasmablasts, and memory B cell populations. (**A**) Gating methodology for identifying cytokine-producing plasma cells and plasmablasts. Subsequent panels demonstrate: (**B**) Phenotypic characterization of splenic CD19^+^CD138^−^ B cell subsets; (**C**) Evaluation of CD19^+^CD138^+^ plasmablast populations in spleen; (**D**) Identification protocol for CD19^−^CD138^+^ plasma cells within splenic tissue. (**E**) Flow cytometry visualization strategy for memory B cell detection, with subsequent quantification of (**F**) IgA-expressing CD19^+^IgD^−^ memory B cells and (**G**) IgG-positive CD19^+^IgD^−^ memory B cell subpopulations. * *p* < 0.05, ** *p* < 0.01, *** *p* < 0.001, ns, not significant.

**Figure 6 vaccines-13-00266-f006:**
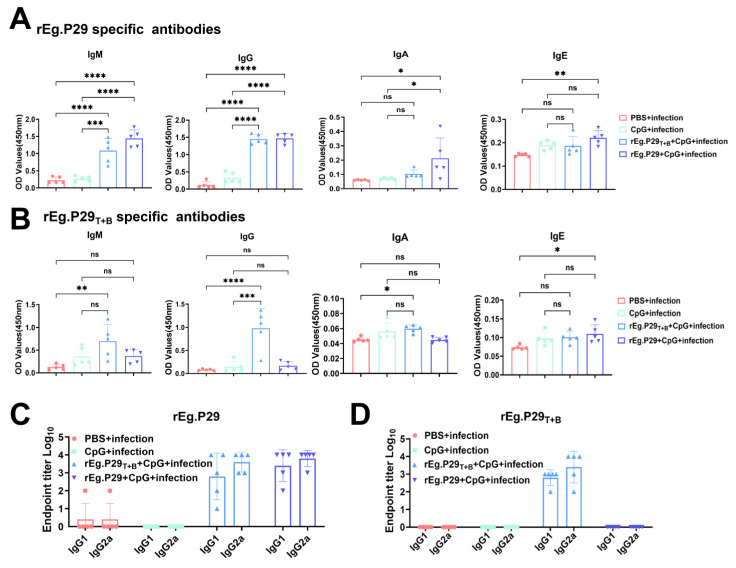
The specific antibody response was tested by ELISA. (**A**,**B**) After the mice were vaccinated by abdominal subcutaneous and infected, the specific rEg.P29 and rEg.P29_T+B_ antibody level was detected by ELISA. (**C**,**D**) End-point titration was used to detect specific IgG1 and IgG2a antibody levels. * *p* < 0.05, ** *p* < 0.01, *** *p* < 0.001, **** *p* < 0.0001, ns, not significant.

**Table 1 vaccines-13-00266-t001:** Monoclonal antibodies for FACS analysis.

Species	Antigen	Fluorochrom	Clone	Supplier	Cat. No.
Mouse	CD19	APC-Cy7	ID3	BD Pharmingen	557655
Mouse	CD138	BV421	281-2	BD Pharmingen	562610
Mouse	IgG	APC	Poly4053	Biolegend	405308
Mouse	IgD	BV786	C10-1	BD Pharmingen	563618
Mouse	IgA	BV605	11-26c.2α	BD Pharmingen	743295
Mouse	CD3	PE-CF594	145-2C11	BD Pharmingen	562286
Mouse	CD4	APC-Cy7	GK1.5	BD Pharmingen	552051
Mouse	CD8	Pacific Blue	53-6.7	BD Pharmingen	558106
Mouse	IFN-γ	FITC	XMG1.3	BD Pharmingen	554411
Mouse	IL-2	PE	JES6-5H4	BD Pharmingen	554428
Mouse	CD44	APC	IM7	BD Pharmingen	561862
Mouse	CD25	PE	PC61	BD Pharmingen	553866
Mouse	CD127	FITC	A7R34	Biolegend	135007
Mouse	CD107a	APC	1D4B	Biolegend	121614

## Data Availability

The datasets presented in this study can be found in online repositories.

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
