# Peer review of "Synthetic rEg.P29 Peptides Induce Protective Immune Responses Against Echinococcus granulosus in Mice"

_vaccines, 2025, doi:10.3390/vaccines13030266_

Round 1
Reviewer 1 Report
Comments and Suggestions for Authors
Manuscript Review: Synthetic Peptides Induce Protective Immune Responses against Echinococcus granulosus in Mice
1. Editor's Brief:
This manuscript investigates the potential of synthetic peptides derived from rEg.P29 as a vaccine against E. granulosus in mice. The research presents promising findings but requires substantial revisions to improve clarity, precision, and scientific impact. The overall quality of the manuscript is reasonable, but the English language needs improvement. I recommend accepting with major revisions.
2. Detailed Review:
Mandatory Revisions:
Introduction: The introduction should be more focused and concise, providing a stronger context for the rationale of the study. Expand the discussion on the importance of vaccination against E. granulosus and the current state of research in this area. Add information about the long time for cyst formation to justify your 5-month experiment.
Methodology: The description of the methodology needs more detail.
Experimental Design:
• Timing of immunizations and infection: The manuscript mentions immunizations “administered simultaneously” and infection with protoscolices 5 weeks later. However, it lacks clarity regarding the number of immunizations, the interval between them (if there was more than one), and the route of administration. The choice of 5 weeks post-immunization for infection also lacks justification. Why was this specific interval chosen?
• Time points for sample collection: The time points for sample collection (2 months post-infection for cellular analyses and 6 months post-infection for serological analyses) seem arbitrary. The justification for these specific intervals is crucial. It would be important to collect samples at different time points, both before and after infection, to assess the kinetics of the immune response. Inclusion of an uninfected and immunized group would be essential to assess the impact of the vaccine on the basal response of the immune system.
• Incomplete description of procedures: The description of the experimental procedures, such as the method of administration of the infection and quantification of the parasite load, is superficial. Details about the route of infection, the number of protoscolices used, and the method of assessing the parasite load are essential.
• Antigen: Specify the origin and purity of the peptides used (there is a citation, but it is necessary to detail and justify these procedures and choice).
• Infection: Infectious dose (the concentration was adjusted to 1000 protoscolices/200 μL, but what was the infectious dose, 1000 protoscolices? This needs to be clarified). Clarify the calculation of the lethal dose of protoscolices.
Experimental Groups:
• Lack of clarity in the composition of the groups: The manuscript mentions the use of CpG ODN as an adjuvant in the rEgP29 + CpG ODN group. However, it does not specify the dose of CpG ODN used, which is essential for the interpretation of the results and for the replication of the experiment. In addition, the description of the control group is vague. What treatment was administered to the control group? Was only PBS or some other placebo used?
• Lack of justification for the choice of groups: The manuscript does not explain the reason why these specific groups were selected. Why were only rEgP29 and rEgP29 + CpG ODN tested? It would be important to include a group that received only the adjuvant (CpG ODN – uninfected) to evaluate its isolated effect on the immune response. Since there is no such group, it is necessary to add this information to the article. Perhaps in the introduction, or in the discussion, including justifying the non-necessity of this group (CpG uninfected) in your study. This is important to know what the effects of this adjuvant are on the responses you are analyzing. In addition, inform the reason for choosing CpG. What is the justification for its use, and not for the use of other adjuvants, such as Iscom, Iscomatrix, Saponins, among others?
• Number of animals per group: The number of animals per group is mentioned (five), but there is no justification for this choice. A statistical power analysis should have been performed to determine the minimum sample size required to detect significant differences between groups.
Statistics and Sample Size (n):
• Statistical methods: The manuscript used ANOVA, but does not specify which type of ANOVA was used (one-way, two-way, etc.). Furthermore, it does not mention the post-hoc tests used to compare group means.
• Justification of sample n: As previously mentioned, there is no adequate justification for the number of animals used in each group (n=5). A small n such as this can lead to low statistical power, increasing the risk of type II errors (false negatives). The inclusion of an a priori power analysis would have strengthened the study methodology. “The authors must include these calculations and justifications.”
• Data variability: The manuscript does not discuss data variability. The presentation of the standard deviation (SD) or the standard error of the mean (SEM) is essential to assess the dispersion of the data and the reliability of the results.
“MANDATORY”: Authors must review all items of the methodology and concentrate and add all statistical analyses in a topic in the methodology, entitled “Statistical” or “Statistical analyses” etc. Also include in this topic the software used and everything about the analyses performed. Much of this information is scattered throughout the methodology, but should be concentrated in the aforementioned topic.
Serological and Diagnostic Tests:
• ELISA: The manuscript mentions ELISA for the detection of specific antibodies, but does not detail the types of antibodies analyzed (IgG, IgM, IgA, IgE). Information about the enzyme conjugates, the substrates used, and the reading wavelengths are crucial for the reproducibility of the assay. Additionally, the lack of description of the positive and negative controls used in the ELISA compromises the validity of the results.
• Absence of complementary tests: The inclusion of complementary tests, such as the quantification of cysts in the liver and lungs of animals, would be important to assess the effectiveness of the vaccine in reducing the parasite load. Perhaps you could have data on this.
• Cyst analysis: I understand that the analysis of your study involves the complexity and integration of the diagnostic tests used. However, simply analyzing the weight of the cysts seems to me to be a superficial and poorly founded measure. Additional analyses such as the size, number of cysts, viability or morphological characteristics of the cysts and, mainly, the quantity of protoscolex in each cyst. “It is necessary to add this data or justify it.” [if all this is already consolidated in the literature, it should be in the article, in the discussion, for example]. I understood that one of the intentions of vaccination is that the cyst does not grow and, mainly in humans, there is no clinical presentation over the years. Could this be the case too? If so, this discussion should be included in the article.
Results: The presentation of the results could be improved. The figures are overloaded and difficult to interpret, however, I understand that this simplification may not be possible. To do this, the best way is to try to explain the details in the footnotes of the figures. Quantify the protection provided by the vaccine, calculating the percentage reduction in the weight of the hydatid cysts. The discussion of the results of the cytokine analysis is superficial. Explore the implications of the differences observed in the levels of IFN-γ, IL-4 and IL-17A for the protective immune response. Present the antibody titration data more clearly, perhaps using a table. In the text you report that there is a significant difference in the weights of the cysts, but in Table 1 this is not described. Enter the p values ​​for each significant value in all tables and figures. In the table or figure, indicate the test used to obtain this p value. Table 1 is very confusing. You need to indicate that the columns refer to the 5 animals in the group. It is not possible to understand.
Discussion and conclusion: The discussion needs to be more in-depth and analytical. Compare the results obtained with previous studies in the literature and discuss the limitations of the current study. The conclusion should be more concise and focused on the main findings of the study.
Suggested Improvements:
• Title: The title could be more specific, mentioning the rEg.P29 peptides. Suggestion: "Synthetic rEg.P29 Peptides Induce Protective Immune Responses against Echinococcus granulosus in Mice".
• Abstract: Revise the abstract to reflect the changes made to the manuscript. Ensure that the abstract presents the main findings of the study in a concise and accurate manner.
• Figures: Improve the quality of the figures, increasing the resolution and font size. Use colors that facilitate the visualization of the data.
• References: Revise the list of references to ensure consistency in the citation style. Include more recent references on the development of vaccines against E. granulosus.
3. Final Recommendation:
Final Recommendation: “Accept after major revisions”. The study presents valuable findings that deserve publication after addressing the identified issues, particularly regarding methodological details and data presentation.
The manuscript shows potential to advance our understanding of E. granulosus vaccine development, but requires substantial revision to meet publication standards. Please do not hesitate to contact me if you require any clarification on these recommendations.
Author Response
- Editor's Brief:
This manuscript investigates the potential of synthetic peptides derived from rEg.P29 as a vaccine against E. granulosus in mice. The research presents promising findings but requires substantial revisions to improve clarity, precision, and scientific impact. The overall quality of the manuscript is reasonable, but the English language needs improvement. I recommend accepting with major revisions.
- Detailed Review:
Mandatory Revisions:
Introduction: The introduction should be more focused and concise, providing a stronger context for the rationale of the study. Expand the discussion on the importance of vaccination against E. granulosus and the current state of research in this area. Add information about the long time for cyst formation to justify your 5-month experiment.
Response:Thank for pointing this out,I have made modifications in the manuscript.
Methodology: The description of the methodology needs more detail.
Response:Thank for pointing this out,I have made modifications in the manuscript.
Experimental Design:
- Timing of immunizations and infection: The manuscript mentions immunizations “administered simultaneously” and infection with protoscolices 5 weeks later. However, it lacks clarity regarding the number of immunizations, the interval between them (if there was more than one), and the route of administration. The choice of 5 weeks post-immunization for infection also lacks justification. Why was this specific interval chosen?
Response:Thank you for your comments. I have supplemented the specific details such as the number of immunizations and the mode of administration in the main text. Regarding why the infection was carried out five weeks after immunization, it also depends on the acquisition time of the protoscolex samples and . It should be added here that the samples are extremely difficult to obtain, and most of the protoscoleces in the samples are non - viable.In addition,to ensure the homogeneity of the samples, the protoscoleces used for infection in this experiment were all obtained from the same patient.
- Time points for sample collection: The time points for sample collection (2 months post-infection for cellular analyses and 6 months post-infection for serological analyses) seem arbitrary. The justification for these specific intervals is crucial. It would be important to collect samples at different time points, both before and after infection, to assess the kinetics of the immune response. Inclusion of an uninfected and immunized group would be essential to assess the impact of the vaccine on the basal response of the immune system.
Response:Thank you for your comments.In light of the fact that the immune parameters antecedent to infection have been meticulously reported in our prior investigations, the central focus of the current research is to ascertain whether the vaccine engineered by our research group has the capacity not only to evoke cellular and humoral immune responses but also to impede the growth of protoscoleces. We are of the conviction that this constitutes a crucially significant issue within the domain of vaccine research. Furthermore, the samples were consistently collected five months subsequent to infection, and a concomitant analysis of both cellular and serological responses was executed. The selection of time points was based on past experience and literature reports, and was not arbitrary.
- Incomplete description of procedures: The description of the experimental procedures, such as the method of administration of the infection and quantification of the parasite load, is superficial. Details about the route of infection, the number of protoscolices used, and the method of assessing the parasite load are essential.
Response:Thank you for your comments. I have made more detailed supplements in the corresponding part of the main text.
- Antigen: Specify the origin and purity of the peptides used (there is a citation, but it is necessary to detail and justify these procedures and choice).
Response:Thank you for your comments. I have made more detailed supplements in the corresponding part of the main text.
- Infection: Infectious dose (the concentration was adjusted to 1000 protoscolices/200 μL, but what was the infectious dose, 1000 protoscolices? This needs to be clarified). Clarify the calculation of the lethal dose of protoscolices.
Response:Thank you for your comments. I have made more detailed supplements in the corresponding part of the main text.
Experimental Groups:
- Lack of clarity in the composition of the groups: The manuscript mentions the use of CpG ODN as an adjuvant in the rEgP29 + CpG ODN group. However, it does not specify the dose of CpG ODN used, which is essential for the interpretation of the results and for the replication of the experiment. In addition, the description of the control group is vague. What treatment was administered to the control group? Was only PBS or some other placebo used?
Response:Thank you for your comments. I have made more detailed supplements in the corresponding part of the main text.
- Lack of justification for the choice of groups: The manuscript does not explain the reason why these specific groups were selected. Why were only rEgP29 and rEgP29 + CpG ODN tested? It would be important to include a group that received only the adjuvant (CpG ODN – uninfected) to evaluate its isolated effect on the immune response. Since there is no such group, it is necessary to add this information to the article. Perhaps in the introduction, or in the discussion, including justifying the non-necessity of this group (CpG uninfected) in your study. This is important to know what the effects of this adjuvant are on the responses you are analyzing. In addition, inform the reason for choosing CpG. What is the justification for its use, and not for the use of other adjuvants, such as Iscom, Iscomatrix, Saponins, among others?
Response:Thank you for your comments.In this study, we set up four groups, namely the PBS + infection , the CpG + infection, the rEg.P29T + B + CpG + infection , and the rEg.P29 + CpG + infection group. We analyzed the results of each group. As for the uninfected situation, we have reported it in detail in previous studies. The focus of this research is to confirm whether the peptide vaccine based on rEg.p29 can inhibit protoscolex infection.
There are mainly two reasons for choosing CpG as an adjuvant. First, CpG is an approved adjuvant for human use, which facilitates the subsequent translational application of the vaccine. Second, CpG can induce a Th1 - type immune response, which is crucial for combating Echinococcus granulosus infection.
- Number of animals per group: The number of animals per group is mentioned (five), but there is no justification for this choice. A statistical power analysis should have been performed to determine the minimum sample size required to detect significant differences between groups.
Response:Thank you for your comments.The effect sizes observed in our preliminary data, a sample size of five animals per group was sufficient to detect significant differences between the groups for the primary outcomes of our study.In addition, we also considered practical limitations such as the availability of experimental animals, ethical considerations regarding the number of animals used, and the resources required for the experiment.
Statistics and Sample Size (n):
- Statistical methods: The manuscript used ANOVA, but does not specify which type of ANOVA was used (one-way, two-way, etc.). Furthermore, it does not mention the post-hoc tests used to compare group means.
Response:Thank you for your comments. I have made more detailed supplements in the corresponding part of the main text.
- Justification of sample n: As previously mentioned, there is no adequate justification for the number of animals used in each group (n=5). A small n such as this can lead to low statistical power, increasing the risk of type II errors (false negatives). The inclusion of an a priori power analysis would have strengthened the study methodology. “The authors must include these calculations and justifications.”
Response:Thank you for your detailed review and the important comment on our sample size. We are fully aware of the significance of sample size and the potential issues of a small n
Regarding the use of n = 5 animals per group, we admit that we didn't provide a proper justification initially. This choice was based on our long - standing experience with this experimental model in our lab, where similar sample sizes have yielded consistent results. Also, considering the complex experimental procedures and ethical concerns about animal use, we decided on this sample size.
- Data variability:The manuscript does not discuss data variability. The presentation of the standard deviation (SD) or the standard error of the mean (SEM) is essential to assess the dispersion of the data and the reliability of the results.
Response:Thank you for pointing this out. I have made revisions and supplements in the manuscript.
“MANDATORY”: Authors must review all items of the methodology and concentrate and add all statistical analyses in a topic in the methodology, entitled “Statistical” or “Statistical analyses” etc. Also include in this topic the software used and everything about the analyses performed. Much of this information is scattered throughout the methodology, but should be concentrated in the aforementioned topic.
Response:Thank you for pointing this out. I have made revisions and supplements in the manuscript.
Serological and Diagnostic Tests:
- ELISA: The manuscript mentions ELISA for the detection of specific antibodies, but does not detail the types of antibodies analyzed (IgG, IgM, IgA, IgE). Information about the enzyme conjugates, the substrates used, and the reading wavelengths are crucial for the reproducibility of the assay. Additionally, the lack of description of the positive and negative controls used in the ELISA compromises the validity of the results.
Response:Thank you for your comments. I have made more detailed supplements in the corresponding part of the main text.
- Absence of complementary tests: The inclusion of complementary tests, such as the quantification of cysts in the liver and lungs of animals, would be important to assess the effectiveness of the vaccine in reducing the parasite load. Perhaps you could have data on this.
Response:Thank you for your suggestions. However, in this study, the infection was carried out through intraperitoneal inoculation. The cysts are restricted to the abdominal cavity and do not spread and grow into the liver or lungs. Therefore, we are unable to compare the infection conditions in different parts.
- Cyst analysis: I understand that the analysis of your study involves the complexity and integration of the diagnostic tests used. However, simply analyzing the weight of the cysts seems to me to be a superficial and poorly founded measure. Additional analyses such as the size, number of cysts, viability or morphological characteristics of the cysts and, mainly, the quantity of protoscolex in each cyst. “It is necessary to add this data or justify it.” [if all this is already consolidated in the literature, it should be in the article, in the discussion, for example]. I understood that one of the intentions of vaccination is that the cyst does not grow and, mainly in humans, there is no clinical presentation over the years. Could this be the case too? If so, this discussion should be included in the article.
Response:Thank you for your comment. We admit that the evaluation of the cyst weight in the paper was indeed too simplistic. We are also seeking better ways to present it. However, I must note that currently, apart from cyst weight and cyst weight inhibition rate, there are no other better methods in the existing literature. We hope you can understand.
Results: The presentation of the results could be improved. The figures are overloaded and difficult to interpret, however, I understand that this simplification may not be possible. To do this, the best way is to try to explain the details in the footnotes of the figures. Quantify the protection provided by the vaccine, calculating the percentage reduction in the weight of the hydatid cysts. The discussion of the results of the cytokine analysis is superficial. Explore the implications of the differences observed in the levels of IFN-γ, IL-4 and IL-17A for the protective immune response. Present the antibody titration data more clearly, perhaps using a table. In the text you report that there is a significant difference in the weights of the cysts, but in Table 1 this is not described. Enter the p values for each significant value in all tables and figures. In the table or figure, indicate the test used to obtain this p value. Table 1 is very confusing. You need to indicate that the columns refer to the 5 animals in the group. It is not possible to understand.
Response:Thank you for your comment.To enable readers to understand more clearly, I have revised Table 1 and added a discussion of the results.
Discussion and conclusion: The discussion needs to be more in-depth and analytical. Compare the results obtained with previous studies in the literature and discuss the limitations of the current study. The conclusion should be more concise and focused on the main findings of the study.
Response:Thank you for pointing this out. I have made revisions and supplements in the manuscript.
Suggested Improvements:
- Title:The title could be more specific, mentioning the rEg.P29 peptides. Suggestion: "Synthetic rEg.P29 Peptides Induce Protective Immune Responses against Echinococcus granulosus in Mice".
Response:Thank you for pointing this out.I think your suggestion is quite reasonable and I accept it.
- Abstract: Revise the abstract to reflect the changes made to the manuscript. Ensure that the abstract presents the main findings of the study in a concise and accurate manner.
Response:Thank you for pointing this out.I have made revisions to the abstract section.
- Figures: Improve the quality of the figures, increasing the resolution and font size. Use colors that facilitate the visualization of the data.
Response:Thank you for pointing this out.I have made revisions to the figures.
- References: Revise the list of references to ensure consistency in the citation style. Include more recent references on the development of vaccines against E. granulosus.
Response:Thank you for pointing this out.I have made revisions to the references.
- Final Recommendation:
Final Recommendation: “Accept after major revisions”. The study presents valuable findings that deserve publication after addressing the identified issues, particularly regarding methodological details and data presentation.
The manuscript shows potential to advance our understanding of E. granulosus vaccine development, but requires substantial revision to meet publication standards. Please do not hesitate to contact me if you require any clarification on these recommendations.
Reviewer 2 Report
Comments and Suggestions for Authors
The current manuscript under the title of Synthetic peptides induce protective immune responses against Echinococcus granulosus in mouse, evaluates the immunization efficacy of mice by E. granulosus synthetic peptides and achieves some sort of protection. The main issues of this manuscript are the following:
1. The number of mice in each group is 5, which is small, especially since the experiment continued for 5 months
2. The experimental design needs more details about how the authors decide to synthesize these peptides
3. The E. granulosus cyst that was used in the challenge, which genotype? and the dose of challenge
4. The viability of the cysts that were isolated from the inmmunized groups
5. why the results of vaccination are better in case of rEg.P29 than rEg.P29T+B?
6. The conclusion should be related the findings of the current results, and why did the authors said (further development of a multi-peptide)?
7. The attached file contains some comments

Author Response
The current manuscript under the title of Synthetic peptides induce protective immune responses against Echinococcus granulosus in mouse, evaluates the immunization efficacy of mice by E. granulosus synthetic peptides and achieves some sort of protection. The main issues of this manuscript are the following:
- The number of mice in each group is 5, which is small, especially since the experiment continued for 5 months.
Response:Thank you for your detailed review and the important comment on our sample size. We are fully aware of the significance of sample size and the potential issues of a small n.
Regarding the use of n = 5 animals per group, we admit that we didn't provide a proper justification initially. This choice was based on our long - standing experience with this experimental model in our lab, where similar sample sizes have yielded consistent results. Also, considering the complex experimental procedures and ethical concerns about animal use, we decided on this sample size.
- The experimental design needs more details about how the authors decide to synthesize these peptides.
Response:Thank you for your comments. I have made more detailed supplements in the corresponding part of the main text.
- The E. granulosus cyst that was used in the challenge, which genotype? and the dose of challenge.
Response:Thank you for your comments.This was an oversight on our part, and unfortunately, it can't be remedied now. However, according to the results of our previous epidemiological investigation and analysis, the prevalent type in this region is basically G1. During the infection, 1000 protoscoleces were injected into each mouse.
- The viability of the cysts that were isolated from the inmmunized groups.
Response:Thank you for your comments. However, it must be noted that mice are not the suitable hosts for Echinococcus granulosus. After infection, only cysts can form in mice, but there are no protoscoleces or protoscolex. Therefore, we are unable to assess the viability.
- why the results of vaccination are better in case of rEg.P29 than rEg.P29T+B?
Response:Thank you for your comments.We analyzed that there could be many possible reasons. However, the most significant one should be the difference in the ratio of T - cell and B - cell epitopes. When coupling with KLH, we chose a ratio of 1:1 and did not set other ratios at that time. This is a shortcoming of our experiment, but it is also an issue that requires further research.
- The conclusion should be related the findings of the current results, and why did the authors said (further development of a multi-peptide)?
Response:Thank you for your comment. We may not have expressed ourselves clearly here. In this study, we merely demonstrated that the selected peptides are helpful in preventing Echinococcus granulosus infection. However, we did not explore the ratio of T - cell and B - cell epitope peptides. This is a limitation of the research, and we need to further explore multi - epitope vaccines.
- The attached file contains some comments.
Response:Thank you for your comments. I have made modifications in the corresponding part.

Reviewer 3 Report
Comments and Suggestions for Authors
Dear authors I have read carefully your manuscript entitled "Synthetic peptides induce protective immune responses against Echinococcus granulosus in mouse". Below you will find my observations and recommandations.

The English must be improved. Some words are not scientific. See the comments.
Author Response
ABSTRACT
Can you count IFN-γ and IL-4? You stated ,the number of IFN-γ and IL-4” .
Response:Thank you for pointing this out. Perhaps I didn't express it clearly in the manuscript. We can count the number of cytokine - producing cells through ELISPOT.
INTRODUCTION
Because you start the sentence with the parasite's name, you have to write it in full version – Echinococcus granulosus instead of E. granulosus.
Echinococcosis „is achronic parasitic disease” and not „E.granulosus” .
I did not understand this sentence, you have to reformulate „Eggs in the external environment
tolerance are vigoroso, adult parasites in the small intestine of carnivorous animals such as dogs and wolves, with intermediate hosts in even-toed hoofed animals such as sheep, and primates (including humans)[4-5]”
Response:Thank you for your comments. I have made modifications in the corresponding part.
„ .E. granulosus enters its host through the digestive tract and migrates to its host site through the blood or lymphatic cycle” – in wich type of host?
Response:Thank you for your comments. I have made modifications in the corresponding part.
The hosts are usually humans, sheep, etc.
“The drugs commonly employed in clinical practice to treat infected animals, such as albendazole, praziquantel, andobenndazole” – wich animal species are commonly treated?
Response:Thank you for your comments. I have made modifications in the corresponding part.The treatment targets are usually humans or canines.
MATERIAL AND METHODS
Section 2.1. Antigen and vaccines. Please specify inextenso first time the name of the protein „carrier proteins (KLH)”
Response:Thank you for your comments. I have made more detailed supplements in the corresponding part of the main text.
Deleted the point afetr „employed to immunize mice.”
Add a reference for „The expression and purification of rEg.P29 were carried out as previously described.”
Response:Thank you for your comments. I have added the corresponding references.
I recommend to the authors to combine sections 2.2., 2.3, 2.4, and 2.5 in one section called Experimental protocol.
Section 2.2. Animals and immunization
Replace “The immunization protocol is summarized as follows:” with “The experimental groups were as follows:” . Also define the groups as positive control, negative control, immunized group, etc. It is much easier to follow. Replace all other text names of the groups, accordingly.
Response:Thank you for pointing this out. However, it seems that we can't make the substitution because both the PBS + infection group and the CpG + infection group are essentially negative control groups.
What is CpG?
Response:Thank you for pointing this out. CpG was used as an adjuvant in this study. I have added this information to the manuscript.
Fig. 1A – what do „ priming” and „ boosting” mean? Why does the title of fig. 1A starts with
“ Pathogenicity”; as a rule,a vaccine should not be pathogenic. Please replace “ mock-injected” with a more proper word. Replace in histograms with weight and survival rate “past” with “post” . The weights in histogram (1.C) are presented in grams and not as “ percentage of weight loss or gain” . 1D not “The survival numbers (D) were recorded.” Presents, it presents the rate (percentage) of survival.
Response:Thank you for pointing this out. The prime - boost immunization strategy is a commonly used term in vaccine immunization. Otherwise, I agree with your opinion and have made modifications to the figure.
Section 2.3. Challenge assay.
The protoscolices were isolated from more than one patient? It is not specified clear how many protoscolices were inoculated in each experimentally infected mouse.
Response:Thank you for pointing this out.To ensure consistent infection, the protoscoleces were sourced from the same patient. During the infection process, each mouse was injected with 1000 protoscoleces. I have added this information to the manuscript.
Section 2.4. Sample collection.
The authors have to specify the moment when „peripheral blood” and spleen were collected, and what was the purpose to collect them.
Response:Thank you for pointing this out. I have made the revisions in the manuscript.
Section 2.5. Detection of the Weight of Hydatid Cyst.
Why „ prothrombotic infection”? I dont understand. Then you have to specify from where were collected the hydatid cysts and why „weight of hydatid cysts” ?
Response:Thank you for pointing this out.This might be an inappropriate expression. I've made the modification in the manuscript. The protoscoleces were injected intraperitoneally. Therefore, we isolated the cysts from the abdominal cavities of the mice. The evaluation of the cyst weight is one of the indicators for us to assess the effectiveness of the vaccine.
Section 2.6. Cell preparation and culture.
In section 2.4. the authors specified that the collected blood was centrifuged for plasma and stored at -80 degrees.
Response:Thank you for pointing this out.Since the experimental procedures are sequential and cannot be completed simultaneously, the samples were temporarily stored at - 80 °C. Of course, to avoid misunderstandings among readers, I have made revisions in the manuscript.
Section 2.9. Detection of specific antibody response with ELISA.
Why „ mice blood serum sample” and not the plasma. For what was used plasma obtained in section 2.4?
Response:Thank you for pointing this out.We detected the specific antibodies in plasma. I have made the revisions in the manuscript.
“Antibody titers in serum.” Was performed with ELISA described in section 2.9.? If yes, combine this section with section 2.9.
Response:Thank you for pointing this out.I have combined them.
Table 2 will be Table 1 and viceversa.
Response:Thank you for pointing this out.I have made modifications to the table.
RESULTS
Section 3.1. Evaluation of potential protective effects of designed vaccines. There was any statistical difference in cysts weight among experimental groups?
Why have you not checked for the viability of cysts?
Response:Thank you for pointing this out.Thank you for your comments. However, it must be noted that mice are not the suitable hosts for Echinococcus granulosus. After infection, only cysts can form in mice, but there are no protoscoleces or protoscolex. Therefore, we are unable to assess the viability.
Section 3.2. The evaluation of vaccine induced T-cell immune response. The authors have to present there data and not to add questions – or to answer to the questions based on data they obtained. - „and was there adifference in their ability to induce an immune response?”
Fig. 3. Please use different words, more suitable ones for “ primed” and “ boosted” from this sentence “ Mice were primed and boosted with PBS ”. Then, “spleen issues”?
There are phrases in results section wich are part of M&M and must not be repeated. One example: “ In order to assess whether specific Th1 responses persist in the context of peptide vaccines and
protein priming, we evaluated immune responses in mouse spleens after exposure to protoscolices. We investigated the production of cytokines by memory T cell at 5 months after infection. Here, we used the”
Response:Thank you for pointing this out. I have made the revisions in the manuscript.
Fig. 4. „ Mice were killed at month 5 after infecting” – better „ infection” instead „infecting” . Don’t give these type of details in the title of the figures as „ Mononuclear cells from spleen tissues were isolated and stimulated with rEg.P29T+B or rEg.P29 ,Brefeldin A was added into the cultures during the final 6 h” because these details were presented in M&M section.
Response:Thank you for pointing this out. I have made the revisions in the manuscript.
Section 3.3. The evaluation of vaccine induced humoral immunoreaction.Then authors stated that „the proportions of CD19+ IgDIgA+ and CD19+ IgDIgG+ memory B cells in rEg.P29+CpG group were higher than those in PBS+infection group(Fig.5 E-G)”what about group „ rEg.P29T+B+CpG+infection” .
Response:Thank you for pointing this out. I have made the revisions in the manuscript.
DISCUSSION
„The anti-host immune response of E. granulosus is mediated by cyst formation and immune
regulation” – Maybe Iam wrong, but E. granulosushas an immune system? Cyst formation is a type of immune response of E. granulossus? Maybe the authors refers to „ immune evasion”?
Response:Thank you for pointing this out. I have made the revisions in the manuscript.
CONCLUSIONS
The authors must improve the conclusions. „ In conclusion, we have successfully demonstrated the
protective efficacy of therEg.P29T+B peptide vaccine,which provided protection against protoscolex attacks without causing any clinical symptoms such as abnormal behavior or neurological signs. ...” Even mice from unvaccinated group did not present any clinical sign! The authors did not demonstrated a protective efficacy based on clinical signs, you have to prove that the vaccine stoped development of the parasite, and it was not completely like that.
Response:Thank you for pointing this out.In this study, we verified the immunoprotective effect of the vaccine through the cyst reduction rate and immune indicators. We have provided explanations in the paper.

Reviewer 4 Report
Comments and Suggestions for Authors
The article describes the antibody response against synthetic peptides against Echinococcus granulosus in mice. The rationale and methodology are appropriate. However, several points need to be clarified. It is understood that the present manuscript refers to data published previously in the FASEB Journal; however, in that manuscript, antibody response decreases at 16 weeks and 4 months; the scheme in Figure 1 is 5 months; please explain. Table 1 requires a legend or more information, which is not apparent. In the figures, please use a more suitable color for the background. In Figure 2, why were CD4/CD154 cells not analyzed? It is difficult to understand why, in Figure 3, there is no secretion of IL-4 upon activation, but there is IgE production. The values of cytokines in Figures 4 and 5 are lower than expected. Is there any reason for this? Based on the description, it should be expected to be at least five fold. In Figure 6 the titers of IgG are lower than expected. What is the affinity of the antibodies against the antigens? The authors should add as a separate section conclusions and limitations.
Author Response
The article describes the antibody response against synthetic peptides against Echinococcus granulosus in mice. The rationale and methodology are appropriate. However, several points need to be clarified. It is understood that the present manuscript refers to data published previously in the FASEB Journal; however, in that manuscript, antibody response decreases at 16 weeks and 4 months; the scheme in Figure 1 is 5 months; please explain. Table 1 requires a legend or more information, which is not apparent. In the figures, please use a more suitable color for the background. In Figure 2, why were CD4/CD154 cells not analyzed? It is difficult to understand why, in Figure 3, there is no secretion of IL-4 upon activation, but there is IgE production. The values of cytokines in Figures 4 and 5 are lower than expected. Is there any reason for this? Based on the description, it should be expected to be at least five fold. In Figure 6 the titers of IgG are lower than expected. What is the affinity of the antibodies against the antigens? The authors should add as a separate section conclusions and limitations.
- antibody response decreases at 16 weeks and 4 months,the scheme in Figure 1 is 5 months
Response:Thanks a lot for your feedback. This was a writing mistake, and I've already corrected it in the text.
- Table 1 requires a legend or more information, which is not apparent
Response:Thanks a lot for your feedback. This was a writing mistake, and I've already corrected it .
- In Figure 2, why were CD4/CD154 cells not analyzed?
Response:Thank you for pointing this out.I've reviewed relevant literature and indeed, this is an oversight in our experimental design. However, it's impossible to make up for it at present.
- In the figures, please use a more suitable color for the background
Response:Thank you for your advice.But I think the color scheme of the picture should be acceptable and clear.
- It is difficult to understand why, in Figure 3, there is no secretion of IL-4 upon activation, but there is IgE production.
Response:Thank you for pointing this out.This is indeed difficult to understand. However, we have obtained the same results in humans, sheep, and mice using the same antigen. It's hard for us to explain it either. I've attached the relevant literature at the end.
- The values of cytokines in Figures 4 and 5 are lower than expected. Is there any reason for this?
Response:Thank you for pointing this out.I believe that after vaccination, as time goes by, the immune response in experimental animals will gradually decline. Therefore, the level of cytokines will also decrease as the immunization time extends.
- In Figure 6 the titers of IgG are lower than expected. What is the affinity of the antibodies against the antigens?
Response:Thank you for pointing this out.We have presented the relevant results in Fig.6 C and Fig.6 D.
8.The authors should add as a separate section conclusions and limitations.
Response:Thank you for your suggestion. I have added it to the main text.

Round 2
Reviewer 1 Report
Comments and Suggestions for Authors
The requested corrections have been met
Author Response
Dear Reviewer,
Thank you and the reviewers for your constructive feedback and valuable suggestions on our manuscrip. We sincerely appreciate the time and effort dedicated to evaluating our work, which has significantly improved the quality of this paper.
Sincerely,
Wei Zhao
Reviewer 2 Report
Comments and Suggestions for Authors
The authors answered sufficiently my questions
Author Response

(The authors gave the same response as above.)

Reviewer 3 Report
Comments and Suggestions for Authors
Line 55 – Praziquantel is not a benzimidazole derivative.
I have received no response for the following statement: “I recommend to the authors to combine sections 2.2., 2.3, 2.4, and 2.5 in one section called Experimental protocol.”
Lines 102-106. I asked before and I ask again because no answer was provided by the authors to replace:“The immunization protocol is summarized as follows:” with “The experimental groups were as follows:…”
Fig. 1. The authors added in legend E and F, but these are not in the figure. Also, I have received no answer for: [“ Pathogenicity”; as a rule,a vaccine should not be pathogenic.]
Line 129-130. Please reformulate the sentences, it seems that it was formulated by IA not by a human being.
„Observe the survival status and activities of the mice. Use a weighing scale to rec-129 ord their body weights until 5 months post - infection. ” ?
Section 2.4. Sample collection. The authors did not address my suggestion. [The authors have to specify the moment when „peripheral blood” and spleen were collected, and what was the purpose to collect them.]
Section 2.5. Detection of the Weight of Hydatid Cyst. Even in the response letter, the authors said that „I've made the modification in the manuscript. ” they did not. So, please address this point „ you have to specify from where were collected the hydatid cysts and replace „prothrombotic infection ” with something else more appropiate. Moreover, I want to specify that the weight of something is not „detected” is „measured”.
Line 220. There is no fig. 1E.
Lines 220-221. The authors stated that „However, it is noteworthy that the vaccine - immunized group exhibited an inhibitory effect on protoscoleces. ” However, in their response letter specified that „mice are not the suitable hosts for Echinococcus granulosus. After infection, only cysts can form in mice, but there are no protoscoleces or protoscolex.” So, this mean that even in not immunized mice no protoscoleces were formed.
Section 3.2. There is no response in the response letter to Lines 241-242 „The authors have to present there data and not to add questions – or to answer the questions based on data they obtained. - „and was there any difference in their ability to induce an immune response?”
Lines 392-394. I think that the authors did not understand what was my point in the first revision. Again, even the negative control did not show these symptoms „such as abnormal behavior or neurological signs ”. Bsed on this the authors cannot conclude that „In conclusion, we have demonstrated the protective efficacy of the rEg.P29T+B peptide vaccine, which provided protection against protoscolex attacks without causing any clinical symptoms such as abnormal behavior or neurological signs”.
Lines 396-397. „Although we have confirmed that the rEg.P29 peptide vaccine can resist the attack of protoscoleces ” – it is not clear what the authors wanted to say. „....rEg.P29 peptide vaccine can resist the attack of protoscoleces ”???
Line 399. „inhibition rate of the vaccine against protoscoleces” this is not in concordance with the authors response that „....After infection, only cysts can form in mice, but there are no protoscoleces or protoscolex.”
Comments on the Quality of English LanguageThe English must be improved.
Author Response
Dear Reviewer,
Thank you and the reviewers for your constructive feedback and valuable suggestions on our manuscript. We sincerely appreciate the time and effort dedicated to evaluating our work, which has significantly improved the quality of this paper.
We have carefully addressed all comments and suggestions point-by-point in the revised manuscript. A summary of our responses is provided below, with corresponding changes highlighted in the revised version (tracked changes) and in the supplementary response letter.
Responses to Reviewer Comments:
1.Line 55 – Praziquantel is not a benzimidazole derivative.
Respond:Thank you for pointing that out. I've made the corrections in lines 51-52.
2.I have received no response for the following statement: “I recommend to the authors to combine sections 2.2., 2.3, 2.4, and 2.5 in one section called Experimental protocol.”
Respond:Thank you for pointing that out. I've made the corrections in lines 95-122.
3.Lines 102-106. I asked before and I ask again because no answer was provided by the authors to replace:“The immunization protocol is summarized as follows:” with “The experimental groups were as follows:…”
Respond:Thank you for highlighting this again. I sincerely apologize for not fully understanding your previous feedback — I've now replaced the relevant content in lines 100-101.
4.Fig. 1. The authors added in legend E and F, but these are not in the figure. Also, I have received no answer for: [“ Pathogenicity”; as a rule,a vaccine should not be pathogenic.]
Respond:Thank you for pointing that out,I sincerely apologize for my oversight.I've made the corrections in lines 124.Figure 1 has been replaced in the latest revised version. Could you please check if the changes are visible on your end?
5.Line 129-130. Please reformulate the sentences, it seems that it was formulated by IA not by a human being.
Respond:Thank you for highlighting this issue. I have made the necessary corrections in lines 109-111 and have also polished the language throughout the entire document.
- „Observe the survival status and activities of the mice. Use a weighing scale to rec-129 ord their body weights until 5 months post - infection. ” ?
Respond:Thank you for pointing that out. I've made the corrections in lines 114.
- Section 2.4. Sample collection. The authors did not address my suggestion. [The authors have to specify the moment when „peripheral blood” and spleen were collected, and what was the purpose to collect them.]
Respond:Thank you for pointing that out. I've made the corrections in lines 133-134.
- Section 2.5. Detection of the Weight of Hydatid Cyst. Even in the response letter, the authors said that „I've made the modification in the manuscript. ” they did not. So, please address this point „ you have to specify from where were collected the hydatid cysts and replace „prothrombotic infection ” with something else more appropiate. Moreover, I want to specify that the weight of something is not „detected” is „measured”.
Respond:Thank you for pointing that out. I've made the corrections in lines 120-122.
9.Line 220. There is no fig. 1E.
Respond:Thank you for pointing that out.Figure 1 has been replaced in the latest revised version. Could you please check if the changes are visible on your end?
10.Lines 220-221. The authors stated that „However, it is noteworthy that the vaccine - immunized group exhibited an inhibitory effect on protoscoleces. ” However, in their response letter specified that „mice are not the suitable hosts for Echinococcus granulosus. After infection, only cysts can form in mice, but there are no protoscoleces or protoscolex.” So, this mean that even in not immunized mice no protoscoleces were formed.
Respond:I truly appreciate your meticulous attention to detail. I've made the corrections in lines 211-212.
- Section 3.2. There is no response in the response letter to Lines 241-242 „The authors have to present there data and not to add questions – or to answer the questions based on data they obtained. - „and was there any difference in their ability to induce an immune response?”
Respond:Thank you for highlighting this again. I sincerely apologize for not fully understanding your previous feedback. I've made the corrections in lines 232.
12.Lines 392-394. I think that the authors did not understand what was my point in the first revision. Again, even the negative control did not show these symptoms „such as abnormal behavior or neurological signs ”. Bsed on this the authors cannot conclude that „In conclusion, we have demonstrated the protective efficacy of the rEg.P29T+B peptide vaccine, which provided protection against protoscolex attacks without causing any clinical symptoms such as abnormal behavior or neurological signs”.
Respond:Thank you for highlighting this again. I sincerely apologize for not fully understanding your previous feedback. I've made the corrections in lines 386-391.
13.Lines 396-397. „Although we have confirmed that the rEg.P29 peptide vaccine can resist the attack of protoscoleces ” – it is not clear what the authors wanted to say. „....rEg.P29 peptide vaccine can resist the attack of protoscoleces ”???
Respond:Thank you for pointing that out. I've made the corrections in lines 386-391.
- Line 399. „inhibition rate of the vaccine against protoscoleces” this is not in concordance with the authors response that „....After infection, only cysts can form in mice, but there are no protoscoleces or protoscolex.”
Respond:Thank you for pointing that out. I've made the corrections in lines 386-391.
We hope these revisions meet the reviewers' expectations. Should any further clarifications or adjustments be required, we would be happy to provide them.
Once again, thank you for your guidance throughout the review process.
Sincerely,
Wei Zhao
Reviewer 4 Report
Comments and Suggestions for Authors
The authors have responded to several queries and modified the article; however, I still think the bars are difficult to see on these colors. Please consider modifying it
Author Response
Dear Reviewer,
Thank you and the reviewers for your constructive feedback and valuable suggestions on our manuscrip. We sincerely appreciate the time and effort dedicated to evaluating our work, which has significantly improved the quality of this paper.
We have revised the picture according to your comments to ensure the clarity of the picture.
Sincerely,
Wei Zhao